# A Multidisciplinary Team Guided Approach to the Management of cT3 Laryngeal Cancer: A Retrospective Analysis of 104 Cases

**DOI:** 10.3390/cancers11050717

**Published:** 2019-05-24

**Authors:** Filippo Marchi, Marta Filauro, Francesco Missale, Giampiero Parrinello, Fabiola Incandela, Almalina Bacigalupo, Stefania Vecchio, Cesare Piazza, Giorgio Peretti

**Affiliations:** 1IRCCS Ospedale Policlinico San Martino, 16132 Genoa, Italy; mfilauro@yahoo.com (M.F.); missale.francesco@gmail.com (F.M.); giampiero.parrinello@gmail.com (G.P.); almalina.bacigalupo@hsanmartino.it (A.B.); stefania.vecchio@hsanmartino.it (S.V.); giorgioperetti18@gmail.com (G.P.); 2Department of Otorhinolaryngology—Head and Neck Surgery, University of Genoa, 16132 Genoa, Italy; 3Department of Otorhinolaryngology, Maxillofacial and Thyroid Surgery, Fondazione IRCCS, National Cancer Institute of Milan, University of Milan, 20133 Milan, Italy; fabiola.incandela@istitutotumori.mi.it (F.I.); cesare.piazza@istitutotumori.mi.it (C.P.); 4Department of Radiation Oncology, University of Genoa, 16132 Genoa, Italy; 5Department of Oncology, University of Genoa, 16132 Genoa, Italy

**Keywords:** laryngeal cancer, laryngeal neoplasm, head and neck cancer, multidisciplinary team, prognosis, laryngo-esophageal disfunction

## Abstract

The optimal treatment for T3 laryngeal carcinoma (LC) is still a matter of debate. Different therapeutic options are available: Transoral laser microsurgery (TLM), open partial horizontal laryngectomies (OPHLs), total laryngectomy (TL), and organ preservation protocols (radiation therapy (RT) or chemo-radiation (CRT)). This study aimed to retrospectively evaluate oncologic outcomes of 104 T3 LCs treated by surgery or non-surgical approaches from January 2011 to December 2016 at a single academic tertiary referral center. Each case was evaluated by a multidisciplinary team (MDT) devoted to the management of head and neck cancers. We divided the cohort into two subgroups: Group A, surgical treatment (TLM, OPHLs, TL) and Group B, non-surgical treatment (RT, CRT). For the entire cohort, two- and five-year overall survival (OS) rates were 83% and 56%, respectively. The two- and five-year disease-free survival (DFS) rates were 75% and 65%, and disease-specific survival rates were 93% and 70%, respectively. The N category was a significant independent prognosticator for OS (*p* = 0.02), whereas Group B was significantly and independently associated with DFS (HR 4.10, *p* = 0.006). Analyzing laryngo-esophageal dysfunction-free survival as an outcome, it was found that this was significantly lower in higher N categories (*p* = 0.04) and in cases that underwent non-surgical treatments (*p* = 0.002). Optimization of oncologic outcomes in T3 LCs may be obtained only by a comprehensive MDT approach, considering that different treatment options have heterogenous toxicity profiles and indications.

## 1. Introduction

Retrospective data from the United States National Cancer Data Base have recently demonstrated that overall survival (OS) for patients with laryngeal cancer (LC) has modestly decreased, whereas in the same time span it has improved for most other cancer locations [1,2]. One of the main drawbacks in the management of LC is that the standard of care for intermediate-advanced disease is still a matter of international debate. Both organ preservation protocols, chemoradiation (CRT)/radiation (RT) alone and surgery in the form of transoral laser microsurgery (TLM), open partial horizontal laryngectomies (OPHLs), as well as total laryngectomy (TL) claim to offer good oncological results. However, in the field of head and neck oncology, purely oncologic results are frequently counterbalanced by considerations of functional outcomes, side effects, and residual quality of life.

Moreover, cT3 LCs comprise different types of lesions that have heterogenous biologic behavior and different possible patterns of failure. In the latest version of the 8th Edition of the Union for International Cancer Control—American Joint Committee on Cancer (UICC-AJCC) TNM staging system, cT3 is defined as a tumor determining vocal fold and arytenoid fixation, invading the pre-epiglottic (PES) and/or paraglottic space (PGS) or minimally infiltrating the laryngeal framework [3]. Arytenoid fixation is considered the best predictive factor for posterior PGS involvement and cricoarytenoid unit (CAU) infiltration [4,5,6,7]. It is worth noting that, when the tumor involves the cricoarytenoid joint and adjacent intrinsic muscles, the recurrent nerve, branches of the inferior laryngeal artery, vein, and related lymphatic vessels may all represent possible pathways for extra-laryngeal tumor spread, causing a high rate of local recurrence and making surgical organ preservation a risky therapeutic option. This recently prompted a proposal of upstaging such posterior lesions to a higher-risk subgroup of cT3 (e.g., called cT3b) [5,6,7].

Different non-surgical organ preservation strategies have been intensively investigated in the last decades. In the 1990s, concomitant CRT became the standard of care for intermediate-advanced LC based on the results reported by Forastiere and coworkers [8], showing substantial evidence to support this approach. In most studies, the primary endpoints were organ preservation (OP) and OS: Among patients with Stage III–IV LC, OP was 88% at two years, with a significant difference in favor of concomitant CRT. OS was apparently excellent (75% at two years) and did not differ significantly according to the treatment protocol (induction chemotherapy followed by RT vs. concomitant CRT vs. RT alone). Despite these good results, in several studies a strong selection bias emerged: Enrolled LCs staged as III–IV often included a heterogeneous group of primary lesions (ranging from cT2 to cT4a) and regional diseases (from cN0 to cN3) [9]. Moreover, the outcomes of glottic and supraglottic LCs were often lumped together, without separate analysis. On top of this, concerning functional outcomes, anatomical organ preservation does not necessarily equate to respect of laryngeal function as demonstrated by the low values of the two-year laryngo-esophageal dysfunction-free survival (LEDFS) [10], defined as patient survival without disease and with no tracheotomy and/or gastrostomy in place [11]. As a consequence, we are still far from achieving international consensus on the best profile of cT3 LC (including patient-related factors) to be optimally treated by non-surgical organ preservation strategies.

The American Society of Clinical Oncology has recently recommended concurrent CRT protocols for the management of cT3 LC, while open-neck organ preservation surgery should be limited to highly selected patients [12]. An international consensus panel composed of experts in treating head and neck cancer recommended that patients diagnosed with cT2-cT3N0 or cN1 glottic or supraglottic tumors and eligible for partial laryngectomies should not be included in organ preservation CRT protocols [11]. By contrast, those with cN2 and cN3 neck will often require adjuvant therapies after surgery and therefore may be considered as ideal candidates for upfront CRT treatment.

This study aims to retrospectively evaluate the appropriateness and consequences of choices made by a head and neck multidisciplinary team (MDT) at a tertiary referral academic center in the management of cT3 LCs, taking into account all the different sources of tumor and patient heterogeneity.

## 2. Materials and Methods

We retrospectively evaluated 518 patients treated for LC from January 2011 to December 2016 at the Department of Otorhinolaryngology—Head and Neck Surgery, IRCCS Ospedale Policlinico San Martino, University of Genoa, Italy. Among those, 104 patients (13 females, 91 males; age range, 47–96 years; mean, 70) affected by untreated cT3 LC met the inclusion criteria for enrolment in the present study. This cohort of patients was composed of supraglottic, glottic, subglottic, or transglottic cT3 LCs treated by surgery (TLM, OPHL, TL) or organ preservation protocols (RT, CRT). Each case was preliminarily discussed by the head and neck MDT of the IRCCS Ospedale Policlinico San Martino University Hospital of Genoa, Italy. The team was composed by head and neck surgeons, radiation and medical oncologists, dedicated radiologists and pathologists, a voice therapist, a speech pathologist, a geriatrician, and a dietician, in order to assess the best therapeutic choice according to the loco-regional extent of disease, comorbidities, overall profile, and preferences. Our decision flow-chart for the management of cT3 LC is shown in Figure 1. All the epidemiological details of the cohort are shown in Table 1.

The diagnostic endoscopic work-up always included a flexible video laryngoscopy under local anesthesia to assess the superficial margins of the lesion under white light (WL), narrow band imaging (NBI, Olympus Medical System Corporation, Tokyo, Japan), and vocal fold/arytenoid mobility. All patients also received intraoperative rigid endoscopy with 0°, 30°, and 70° telescopes under WL and NBI to obtain more information about tumor extension [13].

All patients underwent chest computed tomography (CT). The deep neoplastic extension was evaluated using imaging such as neck CT or magnetic resonance (MR) performed by dedicated radiologists [6,14]. In particular, 101 (97%) patients underwent a preoperative CT scan, while three (3%) an MR. In six (6%) patients, both CT and MR were performed for various reasons (mainly doubts concerning subtle erosion of the inner cortex of the thyroid cartilage). We paid particular attention to laryngeal compartmentalization into an anterior versus posterior PGS by a frontal plane passing through the arytenoid vocal process and perpendicular to the ipsilateral thyroid lamina, as described in the papers by Succo et al. [5] and Del Bon and coworkers [7]. cT3 anterior to such an ideal line were considered “anterior T3”, while those transgressing it in the posterior direction were defined as “posterior T3”. A neck ultrasound (US) with or without fine needle aspiration cytology was routinely performed. Tumors were reclassified according to the 8th Edition of the UICC-AJCC TNM staging system [3]. The T category, Albumin serum level, history of alcohol (or liquor) abuse, and Karnofsky performance status (TALK score) were systematically evaluated to assess the chance of organ preservation and consequently embrace the most appropriate therapeutic choice [15].

We divided the entire cohort of patients into two groups according to the different treatment strategies adopted: Group A (surgical approaches) and Group B (non-surgical treatments). Furthermore, Group A was divided into three subgroups according to the type of surgery received (TLM, OPHL, or TL). TLM and OPHLs were classified according to the corresponding European Laryngological Society (ELS) classifications [16,17,18,19].

All patients underwent TLM using previously described surgical techniques and instrumentation [20]. Transoral re-excision was achieved in cases of deep or more than one superficial positive margins [21,22]. Patients with a persistent tumor after re-excision, perineural invasion, angioembolization, multiple positive lymph nodes, or extranodal extension (ENE) at the final histopathological evaluation were submitted to adjuvant treatment in terms of RT or CRT. Neck dissection (ND) was performed in adherence with the National Comprehensive Cancer Network guidelines for cT3 LC [23].

With regard to Group B, the RT dose, fractions, and delivery technique regimens were as follows: The total equivalent radiation dose was 70 Gy, and a single dose ranged from 2 to 2.12 Gy, delivered in 33–35 fractions with intensity modulated RT (IMRT) or three-dimensional RT (3DRT) techniques with conventional regimen. No accelerated regimen was administered. Patients received concurrent cisplatin (CDDP) at 100 mg/m^2^ every 21 days, or 40 mg/m^2^ every week for seven weeks, or a combination of docetaxel, cisplatin, and 5-fluorouracil (TPF) with one cycle every 21 days, or cetuximab with one cycle every week for eight weeks.

Considering the follow-up policy, we adhered to the ELS guidelines: An endoscopic control schedule was performed every two months during the first two years, every three months during the third year, every six months in the fourth and fifth years, and then annually. Every patient received CT or MR, performed every 4–6 months in the first two years after the treatment even in the absence of suspicious clinical/endoscopic findings [24,25].

No approval from the ethics committee was deemed necessary for this study at our Institution after a formal request was made to the appropriate parties. Each patient signed an informed consent for treatment of personal data for scientific purposes before treatment.

## 3. Statistical Analysis

Qualitative variables were described as absolute and relative frequencies. We considered the following survival endpoints: For OS, the date of death from all causes or date of last consultation for patients alive at the end of the study (censored observations); for disease-free survival (DFS), the date of the first recurrence; for disease-specific survival (DSS), patients who died from unrelated causes were considered as censored observations at the date of death from the disease. Among patients who underwent both surgical and non-surgical OP strategies, LEDFS was calculated by considering as events the date of tracheotomy, TL, gastrostomy, or death from loco-regional disease. Patients who died without loco-regional recurrence were considered as censored observations at the date of death.

Univariate and multivariate survival analysis were performed using the Kaplan–Meier method with a log-rank test (followed by *p* values adjustment with the Benjamini & Hochberg method, for multiple comparisons) [26] and Cox proportional hazard models, respectively. A Schoenfeld residuals test was applied to assess the proportional hazard assumption. Estimates were reported as two- and five-year survival probability and Hazard Ratios (HR) with 95% Confidence Intervals (CI). In all analyses, a two-tailed *p* value < 0.05 was considered significant. R version 3.5.2, SPSS version 23.0 (Chicago, IL, USA), and GraphPad Prism (San Diego, CA, USA) were used for statistical analysis.

## 4. Results

### 4.1. Treatment Characteristics

Among the 104 patients satisfying the inclusion criteria, 66 (63.5%) received surgical treatment (Group A), while 38 (36.5%) underwent non-surgical therapies (Group B).

Group A included 44 (66%) TLM, 11 (17%) OPHL, and 11 (17%) TL. Twenty-seven patients (41%) received ND. Among the 44 patients treated by TLM, 39 were pT3cN0, one was pT3pN0, one was pT3pN1, two were pT3pN2a, and one was pT3pN3b (in fact, one patient previously staged as pN2a according to the 7th Edition of the TNM was restaged as pN3b due to the presence of ENE in the neck specimen). Eight were supraglottic lesions with PES involvement, and 36 were glottic lesions with anterior PGS invasion. Ten glottic lesions were treated by a Type IV cordectomy, 23 by a Type V cordectomy, and three by a Type VI cordectomy [17]. Among the supraglottic tumors, four were treated by a Type III supraglottic endoscopic laryngectomy and four by a Type IV supraglottic endoscopic laryngectomy [18]. One patient treated by TLM received prophylactic ND due to the supraglottic tumor location. Four patients with clinically positive nodes underwent ipsilateral selective ND (levels II–IV).

Five patients underwent OPHL Type I with simultaneous ND: Three were pT3N0 and two were pT3N3b (in fact, two patients previously staged as pN2a according to the 7th Edition of the TNM were restaged as pN3b due to ENE) [19]. All had supraglottic lesions with PES involvement. Only one patient presented limited extension through the posterior PGS, even though with preserved arytenoid mobility. Six patients received OPHL Type II with simultaneous ND: All turned out to be pT3N0 [19]. Four were glottic tumors with anterior PGS involvement, one presented an anterior PGS extension with a suspected thyroid cartilage infiltration at CT, and one had limited extension through the posterior PGS without arytenoid fixation. In this latter case, surgical resection included the ipsilateral arytenoid.

Among the 11 patients who underwent TL and ND, 10 were pT3N0, and only one was pT3N1. Two were transglottic tumors with posterior PGS involvement, arytenoid fixation, and signs of inner thyroid cartilage infiltration at CT. One was a transglottic tumor with posterior PGS involvement, arytenoid fixation, and no signs of cartilage erosion. Two were transglottic tumors with posterior PGS involvement, no arytenoid fixation, and no signs of thyroid cartilage infiltration at imaging. One was a transglottic tumor with posterior PGS involvement, no arytenoid fixation, but with signs of thyroid cartilage erosion at CT. Four patients presented with a transglottic tumor involving the anterior PGS, without arytenoid fixation, and no signs of thyroid cartilage erosion. The last patient had a transglottic lesion involving the anterior PGS, without arytenoid fixation, but with signs of thyroid cartilage erosion.

Twenty-six (39.4%) patients underwent adjuvant treatments. Fifteen (58%) were treated by TLM as upfront treatment, six (23%) by OPHLs, and five (19%) by TL. The adjuvant treatments included RT alone (*n* = 22) or CRT (*n* = 4) for the following reasons: 12 patients for multiple superficial or deep positive margins, three for presence of ENE at the pathological report, and 11 for perineural spread and lympho-vascular invasion.

Group B included 38 patients. There were 24 cT3N0, one cT3N1, one cT3N2a, six cT3N2b, and six cT3N2c. Fifteen patients (39%) were treated by RT alone and 23 (61%) by concurrent CRT. Twenty-two patients were treated by the IMRT technique, 11 by 3DRT, and five by a VMAT approach. Fifteen patients received concurrent CDDP, four TPF, and four cetuximab.

### 4.2. Association of Clinical and Radiological Findings

A strong association was found between the presence of arytenoid fixation and radiological evidence of posterior PGS involvement (*p* < 0.0001), with arytenoid fixation being a predictor of such posterior extension with a specificity of 99%, sensitivity of 50%, positive predictive value of 93%, and negative predictive value of 84%. All cases with arytenoid fixation but one (93%) had posterior PGS involvement, whereas 75 (84%) of the patients with normal arytenoid mobility presented just an anterior PGS involvement (Figure 2).

### 4.3. Groups Comparison Analysis

As the choice of treatment depends on the characteristics of both the patient and the tumor, our analysis confirmed that Groups A and B, as expected, were significantly different with regard to N category (*p* = 0.006), tumor site (*p* = 0.002), presence of arytenoid fixation (*p* = 0.009), and TALK score (*p* = 0.003) (Table 1). On the other hand, no significant differences were observed between Groups A and B in terms of gender and age distributions (Table 1).

### 4.4. Survival Analysis

The median follow-up was 24 months (range, 6–84). At the last follow-up (September 2018), 27 patients had died, 14 of disease progression and 13 from other causes. Three patients were alive with the disease, while the remaining 74 were alive and well with no evidence of the disease.

Twenty-four (23%) of the patients of the entire study had a recurrence. Among Group A, nine (14%) experienced recurrences (seven loco-regional and two at distant sites): All but one of them had received TLM as upfront treatment. They were rescued by OPHLs (*n* = 3), TL (*n* = 2), redo-TLM (*n* = 1) with uni- or bilateral ND, and CRT (*n* = 3). Fifteen (39%) patients of Group B recurred (nine local, two loco-regional, one regional, and three at distant sites), eight after RT alone and seven after CRT. We managed these recurrences with TL (*n* = 4), TLM (*n* = 2) with uni- or bilateral ND, ND alone (*n* = 1), and palliative chemotherapy (*n* = 8).

### 4.5. Univariate Survival Analysis

For the entire cohort, two- and five-year OS rates were 83% and 56%, respectively. A higher N category was significantly associated with worse OS (*p* = 0.005), while no other significant association with OS was found (Figure 3, Table 2). The two- and five-year DFS rates were 75% and 65%, while the corresponding DSS rates were 93% and 70%, respectively. A higher N category (*p* = 0.02 and *p* = 0.02) and Group B (*p* = 0.003 and *p* = 0.016) were significantly associated with worse DFS and DSS (Figure 3, Table 2). Age, gender, T site, and arytenoid fixation were not significantly associated with the outcomes measured (Figure 3, Table 2).

### 4.6. Multivariate Survival Analysis

Multivariate analysis for OS, DFS, and DSS was performed, including as covariates in Cox proportional hazard models, the Group and variables whose proportions were significantly different between the two groups (N category, T site, and arytenoid fixation). A Schoenfeld residuals test was applied, and the proportional hazard assumption was confirmed for all models. The N category was a significant independent prognosticator for OS (*p* = 0.02), whereas being in Group B was significantly and independently associated with DFS (HR 4.10, *p* = 0.006). No variable was associated with different DSS at multivariate analysis.

### 4.7. OP Approaches and LEDFS

Among patients who underwent OP treatments (TLM, OPHL, RT, and CRT), the type of treatment was not associated with different OS (*p* = 0.41) or DSS (*p* = 0.15), but it entailed different two- and five-year DFS, better for OPHL (100% and 100%) compared to TLM (79% and 71%), CRT (65% and 65%), and RT alone (52% and 0%) (*p* = 0.015) (Figure 4). Analyzing LEDFS as an outcome, it was found that it was significantly lower in higher N category tumors (*p* = 0.004) and in cases that underwent non-surgical treatments (Group B, *p* = 0.002) (Table 2, Figure 4). At multivariate analysis, both N category (*p* = 0.04) and Group (*p* = 0.002) were found to be significant independent prognosticators of LEDFS (Table 3).

## 5. Discussion

Regarding the different choices available for the management of T3 LC, each has considerable pros and cons, and the role of a highly performant MDT should be that of balancing them to treat every patient evaluated in the best way possible [27].

As a matter of fact, in the literature, there is a redundancy of specific indications and limitations for each therapeutic aim. For example, optimal laryngeal exposure should be considered of paramount importance when dealing with an anterior T3 through a TLM approach [28,29,30,31]. By contrast, age and comorbidities do not preclude (or even strongly suggest) use of a mini-invasive surgical approach such as TLM, especially considering its rapid operating time and short in-hospital stay, with limited functional sequelae and complications [32,33,34]. Definitely, however, the amount of LC T3 amenable to such a straightforward treatment is relatively limited even in experienced centers [35]. Moreover, it should be the result of a highly selective decision process based on state-of-the-art endoscopic and radiologic compulsory work-up, sometimes to be intraoperatively completed by surgical exploration via the TLM approach (e.g., to discern a subtle inner thyroid cartilage erosion, sometimes undetectable even with the best imaging technique) [36]. Our study demonstrated that such an OP surgical approach presents oncologic outcomes that are comparable (if not better) to those obtained with CRT. Moreover, its minimally-invasive profile is beneficial, especially in terms of function preservation [37] (Figure 4). In fact, if one considers the LEDFS of TLM vs. CRT, the superiority of the former is striking. Considering the wider indications of TLM when compared to those of CRT (since RT alone performed quite poorly in the present study), age and comorbidities contraindicating non-surgical protocols are, per se, strong indications for the management of selected anterior T3 LC through a transoral route [2,34].

The OPHL approach was herein confirmed to be of value for the management of selected unfavorable conditions such as those represented by limited laryngeal exposure or more advanced lesions that are not treatable by a transoral approach, i.e., T3 for inner thyroid cartilage erosion or arytenoid fixation for posterior PGS involvement [38]. Cardiopulmonary and neurologic conditions must also be adequately assessed before embracing such a therapeutic strategy [39]. In fact, to date, no reproducible stratification algorithm has been demonstrated to be effective in predicting the adequate profile of patients to be managed by OPHL, since adequate swallowing recovery may depend on a number of patient-related factors and remains a cumbersome process that can last for months [40]. However, our study demonstrated that after an appropriate tumor and patient selection, oncologic outcomes after OPHLs are excellent and by far superior than all other OP options. In this sense, developing efficient means for selecting the most adequate patients to be submitted to such a procedure beforehand should become a key target of future laryngological studies.

In the present study, we registered a worse DFS (HR 4.10, 95% CI 1.50–11.20; *p* = 0.006) and DSS (HR 3.65, 95% CI 0.99–13.47; *p* = 0.052) in the non-surgical group (Group B). These results are probably due to negative selection bias already mentioned in the recent literature. As a matter of fact, often patients receiving non-surgical treatments are older and affected by the worst subgroups of T3 LC (i.e., those located in the posterior PGS, causing fixation of the ipsilateral hemilarynx or erosion of the inner thyroid cortex, massive subglottic extension, and/or multiple positive nodes).

Non-surgical organ preservation strategies have been traditionally based on CRT as the gold standard [8]. In fact, even the present study confirmed the unacceptably high recurrence rates of RT alone for T3 LC (Figure 4). Such a de-escalation attitude, therefore, should be clearly banned since it is oncologically unsafe [41]. Nevertheless, even CRT presents well-known drawbacks. Apart from its toxicity (different from the surgical one but not less dangerous in terms of potentially fatal complications and sequelae) [42,43], its application and success must rely on strict cooperation between different professional figures like surgeons (for diagnostic work-up, management of airway issues, and follow-up), radiation and medical oncologists, dieticians, and speech pathologists. However, even in the best MDT approach to CRT, it should always be made clear to the patient that such an organ preservation treatment sacrifices at least 20% of oncological outcome in the face of a higher chance of OP (at least compared to upfront TL) [44,45,46]. Moreover, in such a trade-off, it should always be quite clear to the MDT, as well as to the patient, that in case of oncological and/or functional failure, salvage surgery is still feasible but rarely by something less than TL [47,48,49,50,51]. Even so, in case of TL after failed CRT, more sophisticated procedures, frequently encompassing use of pedicled or free flaps, with a higher risk of further oncological failure and undesirable side effects, are needed to avoid salivary fistula, vascular blowout, and esophageal stenosis [52,53,54].

In patients deemed to be suboptimal candidates for organ preservation strategies due to tumor- or comorbidity-related considerations, upfront TL should still be considered as an affordable and extremely reproducible procedure. While definitely unsurpassed in terms of oncological radicality, TL may also be compatible with a reasonable quality of life that, even though inherently subjective, is strongly ameliorated by the systematic application of tracheo-esophageal puncture [55,56].

N status is a relatively infrequent constrain in T3 glottic cancer, while it represents a very important issue when dealing with supraglottic lesions. Moreover, ENE and more than two lymph nodes (indicating the need for postoperative CRT) are even rarer conditions. However, presence of an N+ scenario invariably represents an oncologically negative prognosticator and a strong contraindication to OPHLs and RT alone [27]. On the other hand, TLM, if adequate for management of the T site, even when associated with ND, seems to better tolerate post-actinic sequelae.

## 6. Conclusions

In conclusion, to obtain the best oncological outcomes for T3 LC, it is of paramount importance to discuss each clinical case with a dedicated head and neck MDT. Only such an attitude, in fact, guarantees appropriate evaluation of each subject in his/her complexity and choosing the best treatment modality among those available in the modern laryngological armamentarium, as well as those in radiation and medical oncology. Moreover, we strongly believe that T3 represents the ideal benchmark for ensuring strict cooperation between different figures of the MDT (radiologists, dieticians, speech pathologists, and geriatricians) since it represents the most challenging LC category. As a matter of fact, it is the emblem of a paradigmatic shift from a single treatment of choice to a wide gamma of possible treatments chosen on the base of tumor and patient factors. As shown in this study, when a tailored therapeutic strategy is adequately selected and offered according to the proposed flow-chart, all the possible therapeutic options may turn out to have their field of applicability.

## Figures and Tables

**Figure 1 cancers-11-00717-f001:**
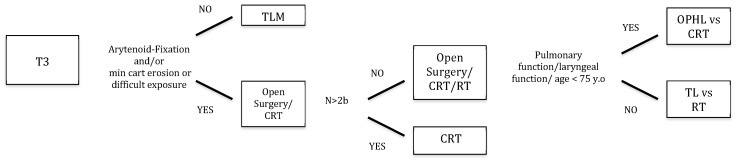
Decision strategy flow-chart. TLM, transoral laser microsurgery; OPHL open partial horizontal laryngectomy; TL, total laryngectomy; RT, radiotherapy; CRT; chemoradiotherapy.

**Figure 2 cancers-11-00717-f002:**
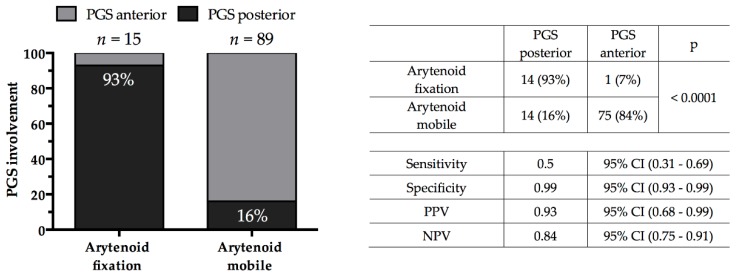
Association between clinical-endoscopic arytenoid mobility (normal vs. fixed) and radiological involvement of the PGS (anterior vs. posterior). *p* value was estimated by Fisher’s exact test. Legend: PGS, paraglottic space; PPV, positive predictive value; NPV, negative predictive value; CI, confidence interval.

**Figure 3 cancers-11-00717-f003:**
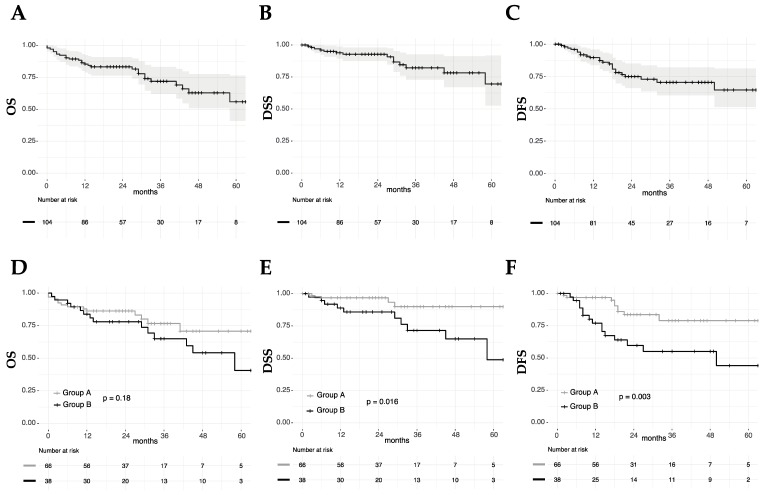
Overall (OS), disease-specific (DSS), and disease-free survival (DFS) for the entire cohort (**A**, **B**, **C**) and for Groups A and B (**D**, **E**, **F**); *p* values estimated by a log-rank test.

**Figure 4 cancers-11-00717-f004:**
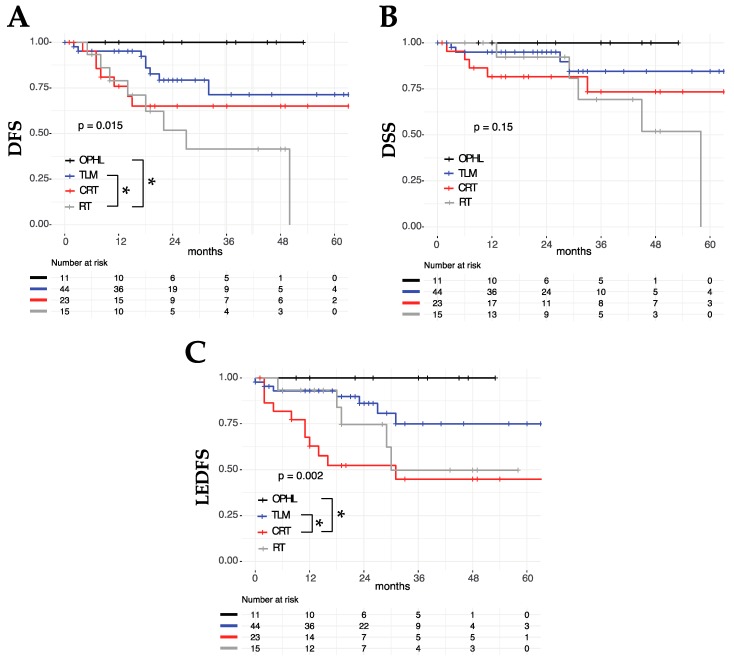
DFS (**A**), DSS (**B**), and LEDFS (**C**) among patients treated by an organ preservation strategy as the first treatment; *p* values estimated by a log-rank test; * *p* < 0.05.

**Table 1 cancers-11-00717-t001:** Demographics and clinical data of the study population (*n* = 104).

Patients Characteristics		All Patients	Group A (Surgical)	Group B (Non-Surgical)	*p*
		*n* (%)	*n* (%)	*n* (%)	
Mean age yr (range yr)		70 (47–96)	69 (47–91)	72 (53–96)	0.08
Gender	Female	13 (12)	9 (14)	4 (11)	0.76
	Male	91 (88)	57 (86)	34 (89)	
N category	N0	83 (80)	59 (89)	24 (63)	0.006
	N1-N2b	12 (11)	4 (6)	8 (21)	
	≥N2c	9 (9)	3 (5)	6 (16)	
Site	Glottic	47 (45)	39 (59)	8 (21)	0.002
	Supraglottic	32 (31)	16 (24)	16 (42)	
	Subglottic	5 (5)	3 (5)	2 (5)	
	Transglottic	20 (19)	8 (12)	12 (32)	
Arytenoid fixation	No	88 (85)	61 (92)	27 (71)	0.009
	Yes	16 (15)	5 (8)	11 (29)	
TALK score	0	17 (16)	13 (20)	4 (10)	0.003
	1	31 (30)	23 (35)	8 (21)	
	2	35 (34)	24 (36)	11 (29)	
	3	21 (20)	6 (9)	15 (40)	
Treatment	TLM	44 (42)	44(66)	-	-
	OPHL	11 (11)	11(17)	-	
	TL	11 (11)	11 (17)	-	
	RT	15 (14)	-	15 (39)	
	CRT	23 (22)	-	23 (61)	

Legend: TALK, Tumor category, Albumin level, Liquor and alcohol consumption, Karnofsky score; TLM, transoral laser microsurgery; OPHL, open partial horizontal laryngectomy; TL, total laryngectomy; RT, radiotherapy; CRT; chemoradiotherapy.

**Table 2 cancers-11-00717-t002:** Univariate survival analysis (*p* values estimated by a log-rank test).

Variables	*n* (%)	OS			DFS			DSS			LEDFS *		
		2-y (%, 95% CI)	5-y (%, 95% CI)	*p*	2-y (%, 95% CI)	5-y (%, 95% CI)	*p*	2-y (%, 95% CI)	5-y (%, 95% CI)	*p*	2-y (%, 95% CI)	5-y (%, 95% CI)	*p*
All	104 (100)	83 (75–89)	56 (37–71)	-	75 (64–83)	65 (48–77)	-	93 (85–96)	70 (46–84)	-	78 (66–85)	66 (52–77)	-
Age													
≥70	54 (52)	83 (69–91)	44 (20–66)	0.39	73 (56–84)	52 (24–74)	0.49	96 (84–99)	59 (25–82)	0.88	77 (60–88)	61 (40–76)	0.92
<70	50 (48)	84 (71–92)	75 (56–87)		76 (60–87)	76 (60–87)		89 (76–95)	85 (67–93)		77 (60–87)	72 (53–84)	
Gender													
F	13 (13)	92 (57–99)	-	0.35	63 (29–85)	-	0.71	92 (57–99)	-	0.91	80 (39–95)	-	0.95
M	91 (88)	82 (72–89)	54 (35–70)		77 (66–85)	65 (47–79)		93 (84–97)	70 (45–85)		77 (65–86)	66 (52–78)	
N category													
N0	83 (80)	90 (81–95)	56 (0–0)	0.005	79 (67–87)	64 (41–80)	0.02	96 (88–99)	69 (34–88)	0.02	82 (70–90)	76 (61–86)	0.004
N1-N2b	12 (11)	61 (25–83)	-		36 (6–69)	-		81 (42–95)	-		64 (23–87)	-	
≥N2c	9 (9)	56 (20–80)	56 (20–80)		71 (26–92)	71 (26–92)		75 (31–93)	75 (31–93)		50 (15–77)	33 (6–66)	
Site													
Glottic	47 (45)	85 (71–92)	57 (32–76)	0.30	75 (58–86)	75 (58–86)	0.98	96 (83–99)	75 (43–91)	0.66	76 (58–86)	71 (51–84)	0.96
Supraglottic	32 (31)	73 (53–86)	51 (25–72)		73 (49–87)	66 (41–83)		89 (68–96)	74 (47–89)		80 (58–91)	56 (28–77)	
Glottic-Subglottic	5 (5)	100%	-		80 (20–97)	-		100%	-		75 (13–96)	-	
Transglottic	20 (19)	90 (65–97)	52 (9–84)		77 (48–91)	58 (18–84)		90 (65–97)	52 (9–84)		78 (46–92)	67 (32–87)	
Arytenoid fixation													
No	88 (85)	85 (76–91)	57 (35–74)	0.36	77 (65–85)	64 (44–78)	0.33	94 (86–97)	69 (42–85)	0.54	80 (67–87)	68 (52–79)	0.17
Yes	16 (15)	73 (44–89)	52 (22–76)		65 (35–84)	65 (35–84)		87 (56–96)	74 (37–92)		61 (27–84)	-	
Group													
A	66 (63)	86 (75–92)	71 (51–84)	0.18	84 (70–92)	79 (62–89)	0.003	97 (88–99)	90 (74–96)	0.016	89 (75–95)	80 (61–91)	0.002
B	38 (37)	78 (61–88)	41 (15–65)		60 (40–75)	44 (20–66)		86 (69–94)	49 (17–75)		61 (42–76)	47 (28–64)	
Type of OP treatment *													
OPHL	11 (11)	90 (47–98)	-	0.41	100%	100%	0.015	100%	100%	0.15	100%	-	0.002
TLM	42 (40)	86 (71–93)	62 (36–80)		79 (61–90)	71 (47–86)		95 (82–99)	85 (62–94)		88 (71–95)	77 (53–89)	
RT-CHT	23 (22)	73 (50–87)	66 (40–83)		65 (40–82)	65 (40–82)		82 (58–93)	73 (46–89)		52 (29–71)	45 (22–65)	
RT	15 (14)	86 (54–96)	0%		51 (22–75)	0%		92 (57–99)	0%		74 (39–91)	-	

Legend: OS, overall survival; DFS, disease-free survival; DSS, disease-specific survival; LEDFS, laryngo-esophageal dysfunction-free survival; OP, organ preservation; OPHL, open partial horizontal laryngectomy; TLM, transoral laser microsurgery; CRT, chemoradiation; RT, radiotherapy; Group A, surgical treatment; Group B, non-surgical treatment; 95% CI, 95% Confidence Interval, - no subjects at risk; * patients that underwent total laryngectomy were excluded.

**Table 3 cancers-11-00717-t003:** Multivariate survival analysis.

**OS**	**B**	**P**	**HR**	**95% CI HR**	
				Inf	Sup
Group B	0.40	0.37	1.50	0.62	3.62
N category		0.02			
N1-2b	1.82	0.01	6.19	1.68	22.81
≥N2c	0.67	0.32	1.95	0.53	7.20
Presence of arytenoid fixation	−0.16	0.80	0.86	0.25	2.90
Site		0.47			
Supraglottic	−0.46	0.40	0.63	0.21	1.86
Glottic-subglottic	−13.76	0.98	0.00	0.00	.
Transglottic	−1.05	0.11	0.35	0.10	1.29
**DSS**	**B**	**P**	**HR**	**95% CI HR**	
				Inf	Sup
Group B	1.29	0.052	3.65	0.99	13.47
N category		0.09			
N1-2b	2.10	0.03	8.16	1.21	54.85
≥N2c	0.99	0.33	2.69	0.36	19.85
Presence of arytenoid fixation	−1.01	0.29	0.37	0.06	2.38
Site		0.80			
Supraglottic	−0.88	0.32	0.41	0.07	2.36
Glottic-subglottic	−13.85	0.98	0.00	0.00	.
Transglottic	−0.32	0.68	0.73	0.16	3.30
**DFS**	**B**	**P**	**HR**	**95% CIHR**	
				Inf	Sup
Group B	1.41	0.006	4.10	1.50	11.20
N category		0.14			
N1-2b	1.50	0.05	4.48	0.98	20.53
≥N2c	0.23	0.78	1.26	0.24	6.67
Presence of arytenoid fixation	−0.69	0.36	0.50	0.12	2.18
Site		0.53			
Supraglottic	−0.77	0.20	0.47	0.14	1.51
Glottic-subglottic	−0.74	0.42	0.48	0.08	2.90
Transglottic	−0.79	0.21	0.45	0.13	1.57
**LEDFS**	**B**	**P**	**HR**	**95% CI HR**	
				Inf	Sup
Group B	1.65	0.002	5.22	1.80	15.16
N category		0.04			
N1-2b	1.70	0.04	5.46	1.11	26.83
≥N2c	1.80	0.02	6.02	1.40	25.90
Presence of arytenoid fixation	−1.22	0.12	0.30	0.06	1.39
Site		0.30			
Supraglottic	−1.16	0.09	0.32	0.08	1.22
Glottic-subglottic	−1.05	0.35	0.35	0.04	3.19
Transglottic	−0.89	0.18	0.41	0.11	1.49

Legend: OS, overall survival; DFS, disease-free survival; DSS, disease-specific survival; LEDFS, laryngo-esophageal dysfunction-free survival; Group A, surgical treatment; Group B, non-surgical treatment; HR, hazard ratio; P, *p* value; B, β coefficient, 95% CI, 95% Confidence Interval.

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
