# Peer review of "A Multidisciplinary Team Guided Approach to the Management of cT3 Laryngeal Cancer: A Retrospective Analysis of 104 Cases"

_cancers, 2019, doi:10.3390/cancers11050717_

Reviewer 1 Report

Dear authors,

Hereby my comments:

How was your power calculation to include the adequate number of patients.

As diagnostic work up how many patients underwent MRI scan as u mentioned.

What was exactly your decision strategy to choice surgery or non-surgery treatment. 

What was your chart flow for your decision strategy depend on clinical and radiological finding.

In group B what was distribution of number of patients in accelerated or conventional schedule.

What was exactly your decision strategy to choice CRT or RT alone.

In your follow up, how many patients received CT and or MRI as u mentioned?  Why was it scheduled every 4-6 months?

In fact you have 3 groups  1- surgery alone 2- surgery + PO(C)RT and 3- (C)RT  with very few number of patients in each group (only 15 in RT alone, 23 in CRT, 26 in surgery + PO(C)RT and 40 in surgery alone group). Again, how calculate you your power for statistical analysis for these groups?

Of your CRT group Nr=23, only 15 received concurrent CDDP.  How did you take it in account  for your survival analysis? Is there a bias due comorbidities in patients who could not receive concurrent CHRT? How was distribution of those patients in your patients groups?

How many events occurs in your survival calculation showed in figure 3, how many patients in each groups?

Your introduction is very long while your discussion is short and not through the point.

Line 284: OP surgical approached present oncologic outcomes that area comparable (if not better) to those obtained with CRT: depend on which data could you prove that and with how many patients? Is there any correction for comorbidity and bias due your retrospectively analysis?

Line 305: your had only 15 patients in your RT alone group.  Depend on which argument and data could you banned de-escalation attitude as you referred?

Line 307: you talked about toxicities in your discussion.  How was that in your study? There is no data presented in your study regard to toxicity.

What was the biggest limitation in your study regard to retrospective character of your study?

Your conclusion is global and does not meet your results and discussion.

Author Response

Answers to Reviewer 1:

1)How was your power calculation to include the adequate number of patients.

Considering DSS as the main oncologic outcome of the study, since previous papers report 5-yr DSS of 60% for patients treated by CRT (Al-Mamgany et al 2012), and ranging from 87% to 97% for those treated by TLM or OPHL (Peretti et al 2016, Succo et al 2018), knowing the ratio between Groups A and B of 0.6 in our center and the accrual and further time available, we should have enrolled at least 53 patients in Group A and 32 in Group B to get a power calculation of 0.80 with an aof 0.05.

However, we herein want to stress how this was not a RCT aimed at demonstrating the superiority of one treatment arm over another but, rather, the opposite. Since T3 are represented by an heterogeneous group of lesions with different patterns of extension (and possible failure), leaving alone the intrinsic diversity of every patient and his/her comorbid profile, the work of an optimal MDT should be that to find, for every clinical scenario, the best therapeutic indication. This paper is just the photograph of what we got applying such a multidisciplinary approach with the tools available today, describing the outcomes thus obtained. 

2) As diagnostic work up how many patients underwent MRI scan as u mentioned.

We added such information in Materials and Methods on lines 111-113 (101 CT scans, 3 MR scans, 6 both CT/MR).

3) What was exactly your decision strategy to choice surgery or non-surgery treatment. 

As detailed in Figure 1, the choice of treatment was done according to a combination of tumor-related and patient-related factors. In particular:

- Patients staged as having a cT3N0 for anterior PGS involvement at imaging and with vocal fold hypomobility but no arytenoid fixation were candidate for TLM; from this group we excluded those with invasion of the posterior PGS and/or inner thyroid cartilage invasion at imaging and/or arytenoid fixation at endoscopy or with inappropriate laryngeal exposure.

- Patients staged as cT3 with arytenoid fixation and/or inner thyroid cartilage erosion with N1-N2a were candidate to both OPHL or TL (according to age, comorbidities, pulmonary function, and pretreatment laryngeal function evaluated as the need for tracheotomy and/or PEG) versus CRT/RT (according to performance status, age, liver, renal, and cardiac function). Both options, when feasible, were discussed with the patient at the end of the MDT, leaving at him/her the final choice.

- Patients staged as cT3 with arytenoid fixation and/or inner thyroid cartilage erosion with N2b-N2c-N3 were candidate to CRT or TL followed by RT (when comorbidities contraindicated concomitant CRT).

4) What was your chart flow for your decision strategy depend on clinical and radiological finding.

See Figure 1 and answer to Question 3.

5) In group B what was distribution of number of patients in accelerated or conventional schedule.

We added such information in Materials and Methods on lines 135-138 (total equivalent radiation dose was 70 Gy, single dose ranged from 2 to 2.12 Gy, delivered in 33-35 fractions with intensity modulated RT (IMRT) or three dimensional RT (3DRT) techniques with conventional regimen. No accelerated regimen was administrated).

6) What was exactly your decision strategy to choice CRT or RT alone.

Choice between CRT versus RT alone was made according to the patient’s performance status, age, liver, renal, and cardiac functions.

7) In your follow up, how many patients received CT and or MRI as u mentioned?  Why was it scheduled every 4-6 months?

We added such details in Materials and Methods on lines 144-146 (every patients received CT or MR, performed every 4-6 months in the first 2 years after the treatment even in the absence of suspicious clinical/endoscopic findings [24,25]). The quoted references explain the reasons why such a time cut-off is considered as the optimal one from our group.

8) In fact you have 3 groups  1- surgery alone 2- surgery + PO(C)RT and 3- (C)RT  with very few number of patients in each group (only 15 in RT alone, 23 in CRT, 26 in surgery + PO(C)RT and 40 in surgery alone group). Again, how calculate you your power for statistical analysis for these groups?

Please see above answer to Question 1.

9) Of your CRT group Nr=23, only 15 received concurrent CDDP.  How did you take it in account  for your survival analysis? Is there a bias due comorbidities in patients who could not receive concurrent CHRT? How was distribution of those patients in your patients groups?

For sure such a bias is present in our study as in every other observational, retrospective study performed outside a RCT, in the real everyday life. Patients selection and their comorbidities definitively impacted heavily on therapeutic choices (either surgical as well as non-surgical) reducing the available tools in those with a worse performance status (such information are detailed in Table 1, referring to the TALK score).

10) How many events occurs in your survival calculation showed in figure 3, how many patients in each groups?

We included subjects at risk in all graphs to clarify the question.

11) Your introduction is very long while your discussion is short and not through the point.

We implemented the discussion according to your observation.

12) Line 284: OP surgical approached present oncologic outcomes that area comparable (if not better) to those obtained with CRT: depend on which data could you prove that and with how many patients? Is there any correction for comorbidity and bias due your retrospectively analysis?

See Figure 4 A,B,C, reference [36], and answer to question 9 for what concerns selection bias.

13) Line 305: your had only 15 patients in your RT alone group.  Depend on which argument and data could you banned de-escalation attitude as you referred?

See Figure 4 A and references [8,40].

14) Line 307: you talked about toxicities in your discussion.  How was that in your study? There is no data presented in your study regard to toxicity.

We added such details in Results on lines 214-223. 

15) What was the biggest limitation in your study regard to retrospective character of your study?

Definitively, the most important bias of the present retrospective study was the different patients’ comorbid and performance status sometimes forcing the MDT to choose suboptimal treatment such as RT alone instead of CRT or TL with PORT instead of OPHL or CRT. However, our aim was not to demonstrate the theoretical superiority of one treatment over another, but to describe different alternatives to be chosen according to different situations in the real life.

16) Your conclusion is global and does not meet your results and discussion

We implemented the conclusion according to your observation.

Reviewer 2 Report

The authors retrospectively analyzed the clinical outcomes of T3 laryngeal carcinoma patients treated with surgical (TLM, OPHLs, TL) or non-surgical (RT, CRT) options which were decided by a multidisciplinary team. Through multivariate analyses, N category was an independent factor of poor OS and LEDFS. Moreover, non-surgical treatment was an independent factor of poor DFS and LEDFS. The study is well designed and the manuscript is refined, giving an impact on head and neck oncologists. Some details should be made clear.

line 31, p value should be 0.04, if the authors talk about the results of multivariate analysis.

(In line 30, the values are from multivariate analysis.)

line 140, Is the approval from the ethics committee unnecessary in Italy? The study uses the personal and clinical data of the patients...

In Table2, there are some blanks (-). Do they mean all patients were dead or censored before 5-y? It should be explained for readers' understanding.

line 233, worse OS seem to be significant for only 2-y. Which does it mean 2-y or 5-y? 

In Figure 3, the authors conducted multiple comparisons, e.g. OPHL vs RT, OPHL vs CRT, or TLM vs CRT in Figure 3C. Was the familywise error rate corrected with Bonferroni method, etc.?

Author Response

Answers to Reviewer 2:

1) line 31, p value should be 0.04, if the authors talk about the results of multivariate analysis.

(In line 30, the values are from multivariate analysis.)

The error has been fixed.

2) line 140, Is the approval from the ethics committee unnecessary in Italy? The study uses the personal and clinical data of the patients...

In our Institution patients routinely sign an informed consent that allows use of their anonymized data for scientific purposes. 

3) In Table2, there are some blanks (-). Do they mean all patients were dead or censored before 5-y? It should be explained for readers' understanding.

Blanks mean that there were no more patients at risk at that time point, either because dead or censored. We added a note for (-) in Table 2.

4) line 233, worse OS seem to be significant for only 2-y. Which does it mean 2-y or 5-y? 

The log-rank test was applied considering all observation time available. Two-yr and 5-yr estimates are the survival probabilities at those specific time-points. We added CI95%to better understand the results. 

5) In Figure 3, the authors conducted multiple comparisons, e.g. OPHL vs RT, OPHL vs CRT, or TLM vs CRT in Figure 3C. Was the familywise error rate corrected with Bonferroni method, etc.?

We carried again the analysis penalizing the p-values by Benjamini Hochberg (BH) method (Benjamini Y. and Hochberg Y, Controlling The False Discovery Rate - A Practical And Powerful Approach To Multiple Testing; November 1995Journal of the Royal Statistical Society. Series B: Methodological 57:289–300; DOI: 10.2307/2346101). All differences showed between groups were still significant, except for OPHL vs. RT for LEDFS as depicted in Figure 4C. 

Reviewer 3 Report

The authors performed a retrospective analysis of 104 cases of cT3 laryngeal cancer patients treated at their institute to evaluate their oncologic outcomes. They also discussed the role of numerous variables in deciding the appropriate treatment recommendations for these patients. The study discusses the importance and benefits of different treatment options at their tertiary referral center. This study adds to the existing evidence of treatment of cT3 laryngeal cancers, taking into consideration that the optimal treatment for this condition is still debatable.

Following are my comments:

Page 2, line 95………”This cohort of patients was composed of …..comorbidities, overall profile, and preferences.” Rephrase this sentence, it is too long.

 Page 3, line 136…………..expand “ELS” to “European Laryngoscopic Society”.

The study evaluated cases from January 2011 to Dec 2016.  How would take recurrent rate into consideration for patients who may have had it after that time range.

Author Response

Answers to Reviewer 3

1) Page 2, line 95………”This cohort of patients was composed of …..comorbidities, overall profile, and preferences.” Rephrase this sentence, it is too long.

We rephrased the sentence according to your suggestions. 

2)  Page 3, line 136…………..expand “ELS” to “European Laryngoscopic Society”.

ELS was previously clarified in line 126-127.

3)The study evaluated cases from January 2011 to Dec 2016.  How would take recurrent rate into consideration for patients who may have had it after that time range.

Time of enrollment was from January 2011 to December 2016. The last follow up was September 2018. Therefore each event happened until September 2018 has been taken into account. See line 246.

Round  2

Reviewer 1 Report

Dear authors,

I thank you for your effort to make this revision.

I have still some remarks:

As a result of DAHANCA 6-7 randomized trial; moderated accelerated radiotherapy significantly improved loco-regional control in patients with laryngeal SCC (except for elderly > 70 years). Would you explain what was the rational for your choice to treat such patients with conventional schedules.

Your reported toxicity profile for radiotherapy and chemoradiation doesn’t  meet the well-known toxicity profile in literature. I advise you to delete this report and focus to the outcomes as your data’s shown not adequately gathered for toxicity study.

Author Response

Answers Reviewer 1:

1) As a result of DAHANCA 6-7 randomized trial; moderated accelerated radiotherapy significantly improved loco-regional control in patients with laryngeal SCC (except for elderly > 70 years). Would you explain what was the rational for your choice to treat such patients with conventional schedules

We would like to thank the Reviewer for his/her critical input in this sense. 

The DAHANCA 6-7 trial compares a standard 5-day fractionation versus a moderately-accelerated 6-day fractionation. In spite of its advantages, however, the latter is not widely applied in our Hospital mainly for practical and logistic reasons. Moreover, due to a lack of improvement in overall survival and increase in acute toxicity, it never gained large acceptance throughout the various Radiation Oncologist Departments of the entire Country.

On the top of this, the NCCN Guidelines and the AIRO (Italian Association of Radiation Oncology) Guidelines for laryngeal SCC (glottic T2 N1 and supraglottic T1-T3 N0-1) suggest either: 
– Fractionation: 66 Gy (2.2 Gy/fraction) to 70 Gy (2.0 Gy/ fraction); daily Monday–Friday in 6–7 weeks

– Concomitant boost accelerated RT: 

·      72 Gy/6 weeks (1.8 Gy/fraction, large field; 1.5 Gy boost as second daily fraction during last 12 treatment days) 

·      66–70 Gy (2.0 Gy/fraction; 6 fractions/wk accelerated)

– Hyperfractionation: 79.2–81.6 Gy/7 weeks (1.2 Gy/fraction, twice daily)

Our treatment policy was therefore in line at least with one of the above mentioned options.

2) Your reported toxicity profile for radiotherapy and chemoradiation doesn’t meet the well-known toxicity profile in literature. I advise you to delete this report and focus to the outcomes as your data’s shown not adequately gathered for toxicity study.

According to your suggestion, we deleted the toxicity profile (previous lines 215-224) from the final version of the manuscript.

Thanks for your revision that we strongly believe contributed to improve the quality of our paper.